# Differential 3′ processing of specific transcripts expands regulatory and protein diversity across neuronal cell types

Saša Jereb[1], Hun-Way Hwang[1†], Eric Van Otterloo[2], Eve-Ellen Govek[3], John J Fak[1], Yuan Yuan[1], Mary E Hatten[3], Robert B Darnell[1]*

[1]Laboratory of Molecular Neuro-Oncology and Howard Hughes Medical Institute, The Rockefeller University, New York, United States; [2]Department of Craniofacial Biology, University of Colorado Anschutz Medical Campus, Aurora, United States; [3]Laboratory of Developmental Neurobiology, The Rockefeller University, New York, United States

**Present address:** [†]Department of Pathology, School of Medicine, University of Pittsburgh, Pittsburgh, United States

**Competing interests:** The authors declare that no competing interests exist.

**Abstract** Alternative polyadenylation (APA) regulates mRNA translation, stability, and protein localization. However, it is unclear to what extent APA regulates these processes uniquely in specific cell types. Using a new technique, cTag-PAPERCLIP, we discovered significant differences in APA between the principal types of mouse cerebellar neurons, the Purkinje and granule cells, as well as between proliferating and differentiated granule cells. Transcripts that differed in APA in these comparisons were enriched in key neuronal functions and many differed in coding sequence in addition to 3′UTR length. We characterize *Memo1*, a transcript that shifted from expressing a short 3′UTR isoform to a longer one during granule cell differentiation. We show that *Memo1* regulates granule cell precursor proliferation and that its long 3′UTR isoform is targeted by miR-124, contributing to its downregulation during development. Our findings provide insight into roles for APA in specific cell types and establish a platform for further functional studies.
DOI: https://doi.org/10.7554/eLife.34042.001

## Introduction

Alternative polyadenylation (APA) is a process by which different ends to an mRNA transcript are determined. These alternative mRNA isoforms differ in the length of their regulatory 3′ untranslated region (3′UTR) and in some cases their coding sequence (CDS). While we have a good understanding of the scope of molecular functions determined by alternative 3′UTR regions – APA has been implicated in the regulation of subcellular protein and mRNA localization, translational regulation and mRNA stability (*Berkovits and Mayr, 2015*; *Tian and Manley, 2017*) – we are only beginning to understand the effects of APA on various physiological processes in vivo. Genome-wide changes in APA have been observed during neuronal activation (*Flavell et al., 2008*), T-cell activation (*Sandberg et al., 2008*), oncogenesis (*Mayr and Bartel, 2009*) and development (*Ji et al., 2009*; *Shepard et al., 2011*; *Wang et al., 2013*), suggesting diverse biologic roles for APA.

Examination of specific genes has provided evidence for important roles for APA in embryonic development and synaptic plasticity. For example, during *Drosophila* development, the long 3′UTR isoform of mRNA encoding Polo kinase is expressed in abdominal epidermis precursor cells and is translated with much higher efficiency than the short 3′UTR isoform expressed in the adult epidermis. Because high levels of Polo protein are required for the proliferation of epidermis precursor cells, deletion of the *Polo* distal polyadenylation signal leads to death during development

(*Pinto et al., 2011*). Another example is *Bdnf* mRNA; its two 3'UTR isoforms each have distinct functions in neurons. The long *Bdnf* isoform is localized to dendrites and translated upon neuronal activity, whereas the short isoform is localized to the cell body and is constitutively translated. Mice that lack the long 3'UTR of *Bdnf* exhibit altered dendritic spine morphology and decreased plasticity of dendritic synapses (*An et al., 2008*; *Lau et al., 2010*).

A comprehensive functional understanding of APA in the brain, however, is lacking. Recently, it has been found that mammalian and fly brains express particularly long 3'UTR isoforms compared to other tissues (*Miura et al., 2013*), suggesting that APA may play a particularly important role in neurons. Current methods have not been able to discern the extent of APA diversity across different neuronal types, and how that may contribute to their morphologic and physiologic diversity. Recently, new approaches, like translating ribosome affinity purification (TRAP), have been developed that enable sequencing of mRNA from specific neurons in a cell type-specific manner (*Mellén et al., 2012*; *Sanz et al., 2013*), but they lack the resolution to precisely identify 3'UTR ends. To address this limitation, we recently developed cTag-PAPERCLIP (conditionally-tagged poly (A) binding protein-mediated mRNA 3' end retrieval by crosslinking immunoprecipitation). cTag-PAPERCLIP – which is based on PAPERCLIP (*Hwang et al., 2016*) and CLIP (*Licatalosi et al., 2008*; *Ule et al., 2003*) – enables purification and sequencing of 3'UTR ends of polyadenylated transcripts via their interaction with poly-A binding protein cytoplasmic 1 (PABPC1), a protein that binds with high specificity to mRNA poly(A) tails. Purifying 3'UTR ends via PABPC1 immuno-precipitation exhibited less internal priming to A-rich regions other than poly-A tails compared to 3'UTR end sequencing techniques based exclusively on oligo-dT priming (*Hwang et al., 2016*). Another major strength of the CLIP approach is that by covalently crosslinking RNA to protein via ultraviolet light, this method captures direct RNA-protein interactions in situ, allowing stringent immunopurification of physiological interactions from non-specific interactions, which is especially important when purifying mRNA from rare cell populations. cTag-PAPERCLIP was recently used to identify APA switches after inflammatory stimulation of microglia in the brain (*Hwang et al., 2017*).

Here we studied APA in the cerebellum, a cortical region of vertebrate brain that is primarily involved in motor coordination and sensory-motor processing (*Buckner, 2013*), because it is composed of well described cell types that are genetically accessible through Cre-driver lines (*Barski et al., 2000*; *Matei et al., 2005*). Using cTag-PAPERCLIP in combination with the appropriate Cre-driver lines, we studied APA in the two principal types of cerebellar neurons: Purkinje and granule cells, which are functionally and morphologically distinct. Purkinje cells, the sole output neuron of the cerebellar cortex, are large, inhibitory neurons with extensive dendritic arbors and a single axon that projects to the cerebellar nuclei. Granule cells, the most numerous neurons in the mammalian brain, are small interneurons that provide the major excitatory input onto Purkinje cells, and receive mossy fiber inputs onto a small number of dendrites (*Butts et al., 2014*). We also compared APA between granule cell precursors and differentiated granule cells. Granule cell precursors proliferate postnatally in the external granule layer, then exit the cell cycle, extend parallel fiber axons that form synapses with Purkinje cells, and then migrate inward, guided by Bergmann glia, to form the internal granule layer (*Edmondson and Hatten, 1987*; *Solecki et al., 2009*).

Using cTag-PAPERCLIP, we found that Purkinje and granule neurons differentially regulate APA on the same transcripts. We also observed such changes in APA during cerebellar granule cell development, when granule cells transition from mitotic precursors to post-mitotic, differentiated neurons wired within the cerebellar circuitry. Transcripts that differed in APA in these comparisons were enriched in key neuronal functions. Differences in APA between the cell types affected 3'UTR length and in some cases CDS. We analyzed one of the transcripts that changes in 3'UTR length during granule cell development, *Memo1* (mediator of ErbB2-driven cell motility 1), which we show to be involved in regulating the proliferation of granule cell precursors in the developing cerebellum. We demonstrate that, as granule neurons differentiate, *Memo1* acquires a longer 3'UTR harboring a miRNA binding site, providing a mechanism for its downregulation during development. Taken together, our findings demonstrate the potential of single cell-type resolution of APA regulation to reveal quantitative as well as qualitative control of functional diversity in different neuronal populations.

## Results

### cTag-PAPERCLIP maps mRNA 3'UTR ends in specific cells in vivo

To investigate the diversity of 3'UTR ends in distinct neuronal types in the mouse brain we compared 3'UTR isoform expression between cerebellar Purkinje and granule cells using cTag-PAPERCLIP. We previously generated a genetically modified mouse, termed *Pabpc1^cTag^*, that expresses GFP-tagged PABPC1 in a Cre-dependent manner. After Cre recombination, the GFP CDS is inserted into the endogenous *Pabpc1* gene locus just upstream of the stop codon, leading to the expression of the PABPC1-GFP fusion protein. GFP antibodies can then be used to immunoprecipitate polyadenylated transcripts from cells expressing GFP-tagged PABPC1 (*Hwang et al., 2017*) (*Figure 1A*). We bred *Pabpc1^cTag^* mice to *Pcp2-Cre* (*Barski et al., 2000*) or *Neurod1-Cre* mice (*Li et al., 2012*), to isolate polyadenylated transcripts from cerebellar Purkinje or granule neurons, respectively.

We first confirmed by immunostaining of cerebellar sections that PABPC1-GFP was expressed specifically in Purkinje or granule cells in these mice (*Figure 1B*). We then performed cTag-PAPER-CLIP (*Figure 1C*). We dissected cerebella from 8 week old mice, UV-crosslinked RNA to protein within the tissue, immunoprecipitated PABPC1-GFP covalently bound to RNA, and radiolabeled this bound RNA (*Figure 1C*, fourth lane). To verify that we specifically co-immunoprecipitated RNA bound to PABPC1-GFP, we digested RNA to completion, leaving only the RNA segment bound to PABPC1 remaining, and on visualizing the radiolabeled RNA-protein complex, observed a single band corresponding to the size of PABPC1-GFP (*Figure 1C*, third lane).

We sequenced co-immunoprecipitated RNA fragments and identified clusters of reads that were present in all biological replicates to demarcate reproducible binding sites. By these criteria, we identified 10,830 and 12,099 clusters belonging to 8575 and 9411 genes in Purkinje and granule cells, respectively. 27% of genes in Purkinje cells (2,336) and 26% of genes in granule cells (2,451) had multiple robust PABPC1 binding sites, suggesting multiple polyadenylation sites for these genes. The number of reads in each of these clusters was highly correlated across biological replicates (*Figure 1D*), indicating that this method is quantitatively reproducible.

We assessed the ability of cTag-PAPERCLIP to measure 3'UTR isoform abundance by comparing the sum of cTag-PAPERCLIP reads per gene with the mRNA abundance estimated from TRAP-Seq, an alternative method for cell-type specific mRNA profiling (*Mellén et al., 2012*). We found a high correlation between the two methods for both Purkinje cells (R = 0.68, *Figure 1E*) and granule cells (R = 0.7, *Figure 1—figure supplement 1*).

Known marker genes for each cell type were among the most highly expressed genes in the Purkinje cell (*Figure 1E*) and granule cell datasets (*Figure 1—figure supplement 1*), indicating that cTag-PAPERCLIP targeted each cell type specifically. Markers from non-target cell types were ranked significantly lower in cTag-PAPERCLIP data (*Figure 1—figure supplement 2*) compared to TRAP-Seq, indicating that cTag-PAPERCLIP is highly selective at purifying RNA from specific cell types.

Two thirds of cTag-PAPERCLIP clusters mapped to the exact end of 3'UTRs of Ensembl-annotated genes (*Aken et al., 2016*) (*Figure 1F*). Interestingly, we also discovered thousands of new 3'UTR ends, most of which (30–31% of clusters) mapped within the 3'UTRs of Ensembl-annotated genes, but not to the exact annotated ends. Importantly, the majority (87%) of clusters in Purkinje and granule cells harbored a canonical poly(A) signal sequence (AAUAAA or AUUAAA) (*Tian et al., 2005*) within the cluster.

Taken together, these data demonstrate that cTag-PAPERCLIP specifically purifies 3'UTR ends from genetically defined cell types, and can reveal a host of new transcript ends in the brain.

### Cell-type specific APA regulates 3'UTR length and CDS of transcripts expressed in Purkinje and granule cells

We then compared 3'UTR isoform expression between Purkinje and granule cells in our cTag-PAPERCLIP data. We analyzed genes that were expressed in both cell types in order to compare the expression of different isoforms of the same gene across cell types. This analysis revealed extensive isoform diversity: 629 genes expressed different 3'UTR isoforms between the two cell types (FDR < 0.05) (*Supplementary file 1*).

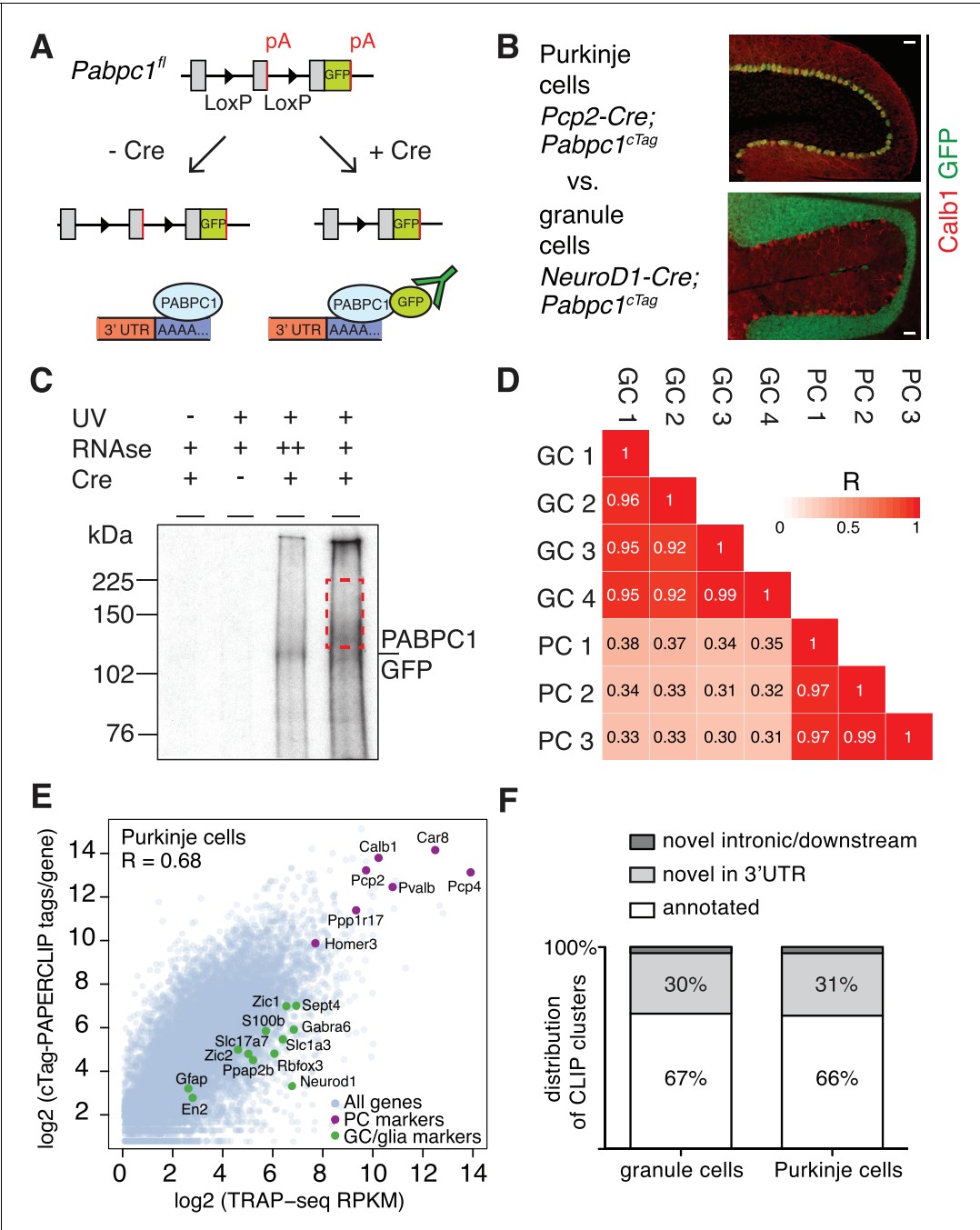

**Figure 1.** cTag-PAPERCLIP identifies 3′UTR isoforms expressed in specific neuronal types. (**A**) Schematic of the cTag-PAPERCLIP approach. Breeding of cTag-PABP mice with Cre-expressing mice restricts expression of PABPC1-GFP to the cells of interest (**B**) Immunostaining of cerebella from mice expressing PABPC1-GFP in Purkinje cells (top) and granule cells (bottom). Calb1: Purkinje cell marker. Scale bars: 50 μm. (**C**) Autoradiogram of radiolabelled RNA cross-linked to PABPC1-GFP purified by immunoprecipitation from granule cells. Red dashed rectangle shows the area of the membrane from which RNA was eluted and sequenced. (**D**) Correlation between the total number of cTag-PAPERCLIP reads per cluster in four biological replicates from granule cells and three biological replicates from Purkinje cells. R: Pearson correlation coefficient. (**E**) Comparison between total uniquely mapped cTag-PAPERCLIP reads per gene and TRAP-Seq Reads Per Kilobase per Million mapped reads (RPKM) per gene from Purkinje cells. R: Pearson correlation coefficient. Purkinje cell markers are highlighted in purple and non-target cell markers are highlighted in green. (**F**) Overlap of cTag-PAPERCLIP clusters that contain reads from three biological replicates from Purkinje cells and at least three biological replicates from granule cells with 3′UTR ends of Ensembl-annotated genes (annotated – clusters that overlap with annotated 3′UTR ends, novel – clusters that do not overlap with annotated 3′UTR ends). The data in panels E and F were derived from analysis of four replicates of cTag-PAPERCLIP on granule cells and three replicates of cTag-PAPERCLIP on Purkinje cells.

*Figure 1 continued on next page*

*Figure 1 continued*

DOI: https://doi.org/10.7554/eLife.34042.002

The following figure supplements are available for figure 1:

**Figure supplement 1.** Comparison between total uniquely mapped cTag-PAPERCLIP reads per gene and TRAP-Seq Reads Per Kilobase per Million mapped reads (RPKM) per gene from granule cells.

DOI: https://doi.org/10.7554/eLife.34042.003

**Figure supplement 2.** Comparison of marker gene ranks from non-target cell types in cTag-PAPERCLIP data and TRAP-Seq data in Purkinje cells.

DOI: https://doi.org/10.7554/eLife.34042.004

Most transcripts differed in 3'UTR length only, without impact on the CDS; among differentially polyadenylated transcripts with two poly(A) sites, 319 or 81% differed in 3'UTR length only (3'UTR-APA, *Figure 2A*). For example, the gene coding for vezatin (*Vezt),* which is involved in neurite outgrowth (*Sanda et al., 2010*), predominantly expressed a shorter 3'UTR isoform in Purkinje cells compared to granule cells (*Figure 2B*). Conversely, *Zfp609*, a neuron-specific transcription factor (*van den Berg et al., 2017*), predominantly expressed a long 3'UTR isoform in Purkinje cells, whereas in granule cells, it expressed both a short and the long 3'UTR isoform. The short 3'UTR isoforms for both *Vezt* and *Zfp609* have not been previously annotated (by Ensembl), and contain a canonical poly(A) signal just upstream of their 3'UTR ends (*Figure 2B*).

Interestingly, in comparing cTag-PAPERCLIP data between Purkinje and granule cells, we also identified transcripts that differed in CDS length (CDS-APA). For example, among transcripts with two poly(A) sites, 19% (76) showed significant differences in CDS-APA between Purkinje and granule (*Figure 2C*). 7 instances of CDS-APA altered the inclusion of phosphorylation sites and 29 instances altered the inclusion of known protein domains, including those with enzymatic activities and those mediating protein-protein interactions (*Supplementary file 2*).

Among transcripts differing in CDS between Purkinje and granule cells was *Ncam1*, which codes for a neural cell adhesion molecule implicated in neurite outgrowth (*Fields and Itoh, 1996*); granule cells mostly expressed the full-length isoform of *Ncam1*, whereas Purkinje cells mostly expressed an isoform coding for a truncated protein (*Figure 2D*). We observed the opposite pattern for *Copg1*, a protein coding for a subunit of the coatomer complex (*Hahn et al., 2000*); granule cells expressed an isoform of *Copg1* coding for a truncated protein in addition to the full-length isoform, whereas Purkinje cells only expressed the full-length isoform (*Figure 2D*).

We also found that previously published TRAP-Seq data from Purkinje and granule cells (*Mellén et al., 2012*) was consistent with the data from cTag-PAPERCLIP for the above mentioned genes (*Vezt, Zfp609, Ncam1* and *Copg1*) (*Figure 2—figure supplement 1A and B*). However, TRAP-Seq was not able to pinpoint 3'UTR ends accurately (*Figure 2—figure supplement 1A*), underscoring the additional resolution of RNA regulation conferred by cTag-PAPERCLIP. To further validate the differences in 3'UTR isoform expression of *Vezt, Zfp609, Ncam1* and *Copg1*, we performed RT-qPCR on RNA purified by TRAP from Purkinje and granule cells. The RT-qPCR data was consistent with cTag-PAPERCLIP, but the latter showed more pronounced differences in 3'UTR isoform expression between the two cell types, presumably because of the more selective purification of RNA (*Figure 2—figure supplement 2*).

We observed that genes involved in the regulation of cell morphology and ion transport were significantly enriched among genes expressing different 3'UTR isoforms in the two cell types (*Supplementary file 3* and *Figure 2E*, which shows the first six most enriched relevant functional categories). The genes included *Sept11,* which has been shown to regulate dendritic branching (*Li et al., 2009*); *Trio*, which has been shown to regulate dendritic branching and synapse function (*Ba et al., 2016*) and *Ube3a*, which has been shown to regulate neuronal excitability (*Judson et al., 2016*). This observation suggests that the striking morphological and physiological differences between granule (excitatory, sparse dendrites) and Purkinje (inhibitory, extensive dendrites) cells may at least in part be mediated by fine-tuning gene expression via APA.

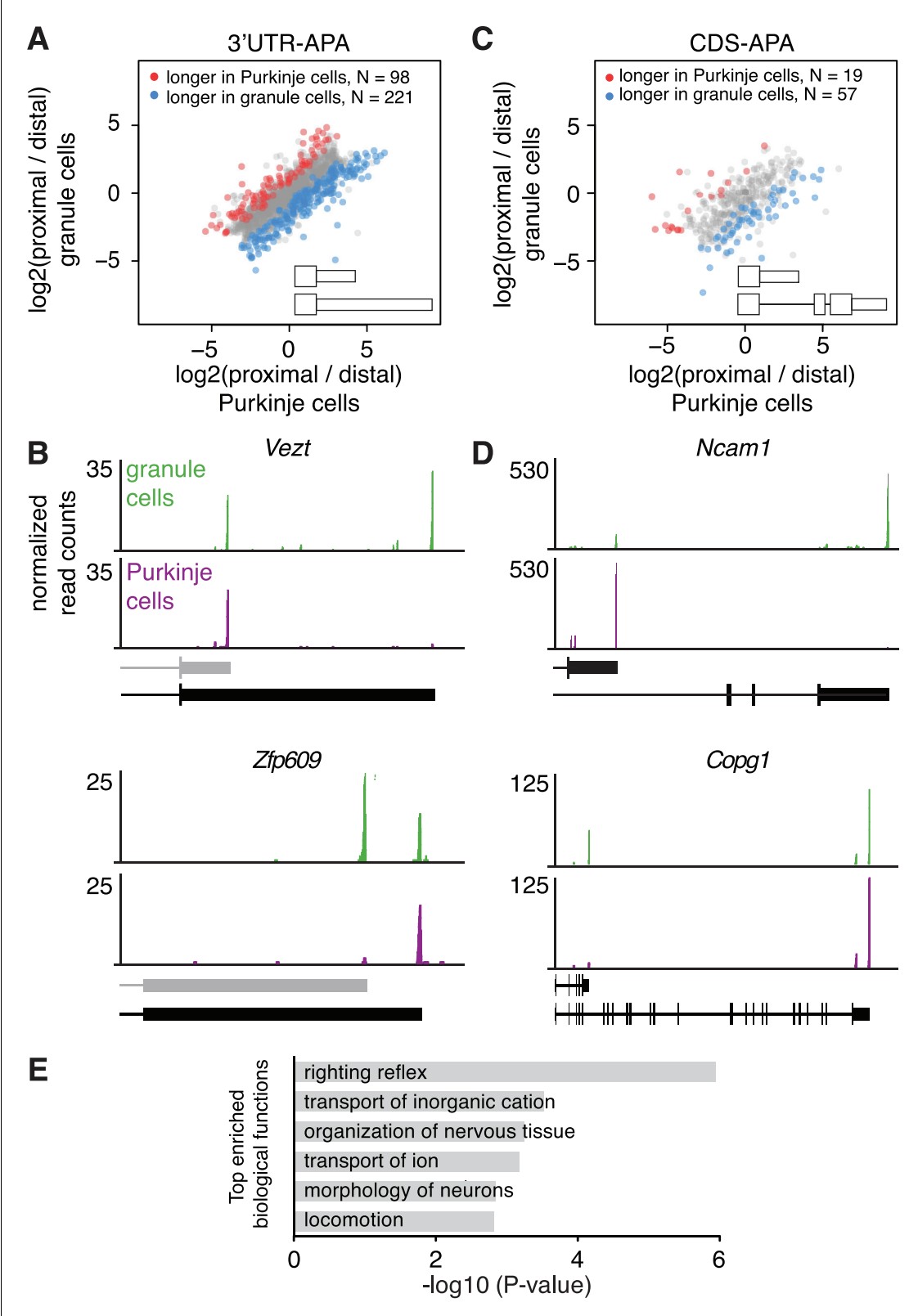

**Figure 2.** Differences in APA between Purkinje and granule cells. (**A**) Scatterplot representing the ratio between the number of cTag-PAPERCLIP reads at the end of the proximal 3'UTR isoform and the number of cTag-PAPERCLIP reads at the end of the distal 3'UTR isoform in Purkinje vs. granule cells. Only genes with two tandem 3'UTR isoforms are shown. Genes showing significantly different ratios (FDR < 0.05) are highlighted in red or blue. (**B**) Examples of cTag-PAPERCLIP data for genes that show a large difference in 3'UTR-APA between Purkinje and granule cells. Black bars represent

*Figure 2 continued on next page*

Figure 2 continued

isoforms annotated by Ensembl, gray ones are (predicted) novel isoforms discovered by cTag-PAPERCLIP. (C) Scatterplot representing the ratio between the number of cTag-PAPERCLIP reads at the end of the proximal 3'UTR isoform and the number of cTag-PAPERCLIP reads at the end of the distal 3'UTR isoform in Purkinje vs. granule cells. Only genes with two 3'UTR isoforms that differ in CDS-APA are shown. Genes showing significantly different ratios (FDR < 0.05) are highlighted in red or blue. (D) Examples of cTag-PAPERCLIP data for genes that show a large difference in CDS-APA between Purkinje and granule cells. (E) Gene ontology analysis of genes showing significant differences in APA between Purkinje and granule cells. Top six relevant functional categories are shown. The data in all panels were derived from analysis of four replicates of cTag-PAPERCLIP on granule cells and three replicates of cTag-PAPERCLIP on Purkinje cells.

DOI: https://doi.org/10.7554/eLife.34042.005

The following figure supplements are available for figure 2:

**Figure supplement 1.** TRAP-Seq data for genes shown in *Figure 2B and D*.

DOI: https://doi.org/10.7554/eLife.34042.006

**Figure supplement 2.** qPCR validation of 3'UTR isoform abundance differences between Purkinje and granule cells.

DOI: https://doi.org/10.7554/eLife.34042.007

## APA alters 3'UTR length and CDS of specific transcripts during granule cell development

To determine if polyadenylation sites are dynamically selected within a given neuronal type as a cell transitions from a mitotic precursor to a post-mitotic, differentiated neuron, we profiled APA in granule cells during development. We analyzed APA at two time points: P0, when granule cell precursors are proliferating in the external granule layer, and P21, when granule cells have completed their migration to the internal granule layer and have made axonal synaptic connections with Purkinje cells, as well as dendritic synaptic connections with mossy fiber and Golgi cell inputs (*Hatten and Heintz, 1995*). We bred cTag-PABP mice with mice expressing Cre under the control of the *Atoh1* promoter (*Matei et al., 2005*) to purify PABP-bound transcripts from both granule cell precursors and differentiated granule neurons. Although *Atoh1* is only expressed in granule cell precursors, their progeny – the differentiated granule cells – inherit the recombined *Pabpc1-GFP* locus.

We first confirmed that PABPC1-GFP is expressed in precursor and differentiated granule neurons by immunostaining on P0 and P21 cerebella (*Figure 3A*). We then performed cTag-PAPERCLIP on dissected cerebellar cortices (discarding deep cerebellar nuclei, which contain non-granule cell *Atoh1*-expressing neurons). We obtained 12,014 and 14,491 clusters of reads, which belonged to 9698 and 10,716 genes expressed in granule cell precursors (from P0 mice) and differentiated granule cells (from P21 mice), respectively. 32% of genes in granule cell precursors (3,104) and 32% of genes in differentiated granule cells (3,436) had multiple clusters.

Comparing cTag-PAPERCLIP data between precursor and differentiated granule cells from P0 and P21 mice, we found 737 genes that significantly changed in APA (FDR < 0.05) (*Supplementary file 4*). 3'UTRs tended to lengthen during granule cell development, which is consistent with RNA expression data from whole developing mouse brains (*Ji et al., 2009*). Specifically, among differentially polyadenylated transcripts with two poly(A) sites in 3'UTRs, 288 out of 354 got longer during granule cell development (*Figure 3B*), For example, *Kpnb1*, which codes for importin subunit β1, whose long 3'UTR isoform has been shown to localize the mRNA to axons (*Perry et al., 2012*), and *Cdk7*, which has been shown to be required for cell cycle progression of neurons (*Abdullah et al., 2016*), both shifted to predominantly expressing a long 3'UTR isoform in differentiated granule cells (*Figure 3C*).

In assessing 3'UTR regulation during granule cell development, we also discovered transcripts where APA altered their CDS. Among differentially polyadenylated transcripts with two poly(A) sites, 98 (22%) displayed changes in CDS. In the majority of cases (69/98), CDS lengthened during granule cell differentiation (*Figure 3D*). Such changes in CDS content indicate that APA can act to modify protein function during granule cell development (*Figure 3D*). For example, 10 of these APA instances altered the inclusion of sites for posttranscriptional modifications and 42 APA instances altered protein domains, such as those with enzymatic, DNA-binding, and protein-binding activity (*Supplementary file 5*).

An example of the effect of APA change during granule cell development is illustrated by examining cTag PAPERCLIP results for the *Sin3b* transcript, which codes for a scaffold subunit of the Sin3-histone deacetylase (HDAC) transcriptional repression complex (*Silverstein and Ekwall, 2005*), and

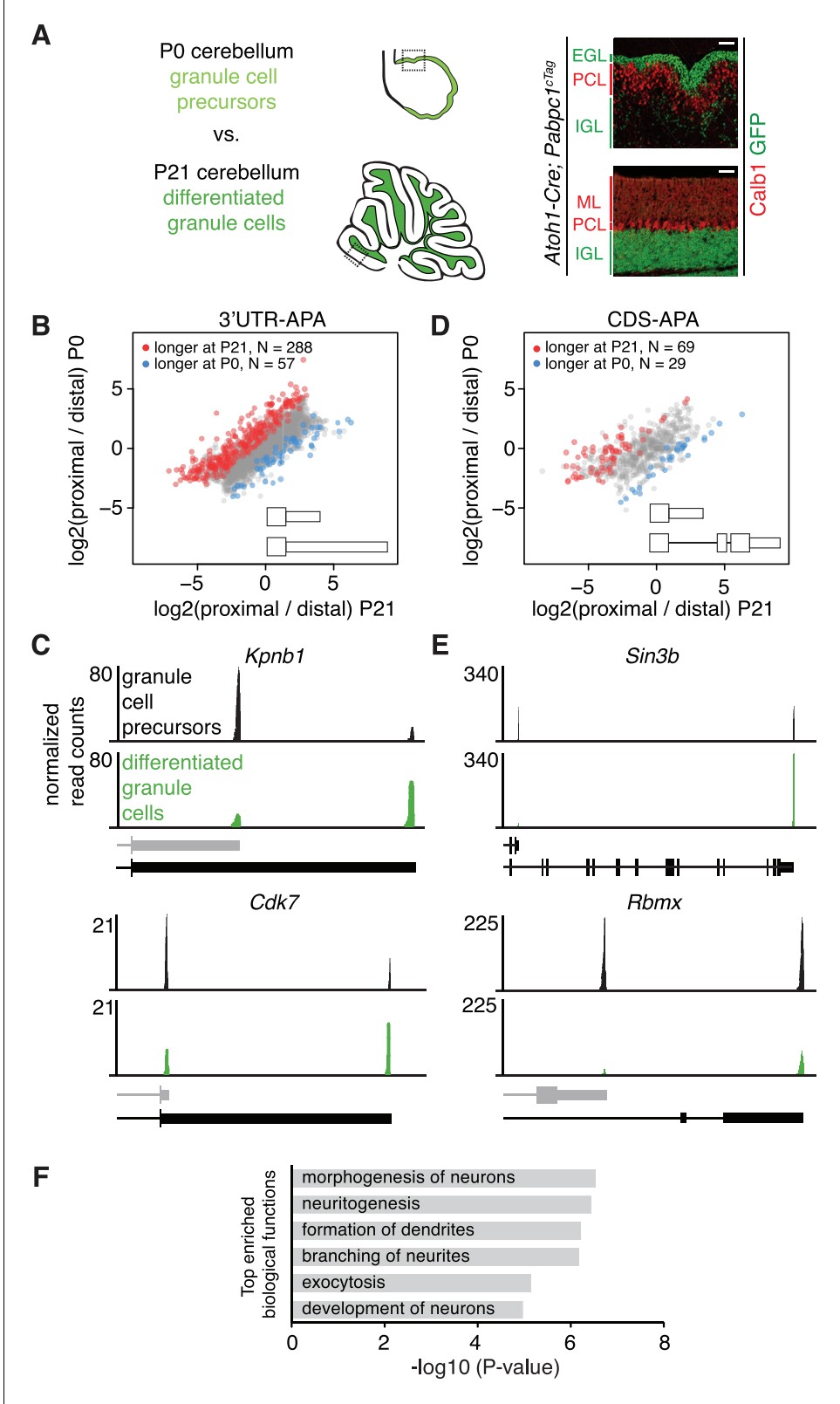

**Figure 3.** Changes in APA during granule cell development. (**A**) Immunostaining showing cell type specific expression of conditionally tagged PABPC1 in P0 and P21 granule cells. EGL – external granule layer, PCL – Purkinje cell layer, IGL – internal granule layer, ML – molecular layer. Calb1: Purkinje cell marker. Scale bars: 50 μm. Dotted square on the schematic of cerebellum at P0 and P21 (left side of the panel) shows the location of the immunostaining on the right. (**B**) Scatterplot representing the ratio between the number of cTag-PAPERCLIP reads at the end of the proximal 3'UTR

*Figure 3 continued on next page*

*Figure 3 continued*

isoform and the number of cTag-PAPERCLIP reads at the end of the distal 3'UTR isoform in P0 vs. P21 granule cells. Only genes with two tandem 3'UTR isoforms are shown. Genes showing significantly different ratios (FDR < 0.05) are highlighted in red or blue. (C) Examples of cTag-PAPERCLIP data for genes that show a large difference in 3'UTR-APA between P0 and P21 granule cells. Black bars represent isoforms annotated by Ensembl, gray ones are (predicted) novel isoforms discovered by cTag-PAPERCLIP. (D) Scatterplot representing the ratio between the number of cTag-PAPERCLIP reads at the end of the proximal 3'UTR isoform and the number of cTag-PAPERCLIP reads at the end of the distal 3'UTR isoform in P0 vs. P21 granule cells. Only genes with two 3'UTR isoforms that differ in CDS-APA are shown. Genes showing significantly different ratios (FDR < 0.05) are highlighted in red or blue. (E) Examples of cTag-PAPERCLIP data for genes that show a large difference in CDS-APA between P0 and P21 granule cells. Black bars represent isoforms annotated by Ensembl, gray (predicted) isoform of *Rbmx* was discovered by cTag-PAPERCLIP. The CDS portion of the last exon in the predicted *Rbmx* isoform was inferred from an annotated isoform not shown in the picture. (F) Gene ontology analysis of genes showing significant differences in APA between P0 and P21 granule cells. Top six relevant functional categories are shown. The data in panels B-F were derived from three replicates of cTag-PAPERCLIP per time point.

DOI: https://doi.org/10.7554/eLife.34042.008

The following figure supplements are available for figure 3:

**Figure supplement 1.** The number of cTag-PAPERCLIP reads per gene is correlated with RNA sequencing reads per kilobase (RPK) per gene.
DOI: https://doi.org/10.7554/eLife.34042.009

**Figure supplement 2.** RNA-seq data for genes shown in *Figure 3C and E*.
DOI: https://doi.org/10.7554/eLife.34042.010

**Figure supplement 3.** qPCR validation of 3'UTR isoform abundance differences between P0 and P21 granule cells.
DOI: https://doi.org/10.7554/eLife.34042.011

**Figure supplement 4.** Distribution of log2 fold changes in ribosome-associated mRNA abundance between P0 and P21 cerebellar granule cells for groups of genes exhibiting significant differences in 3'UTR isoform expression during development.
DOI: https://doi.org/10.7554/eLife.34042.012

which has been shown to be necessary for cell cycle exit and differentiation (*David et al., 2008*). Differentiated granule cells express the long isoform of *Sin3b*, whereas granule cell precursors express both the long isoform and a short isoform lacking the HDAC-interaction domain (*Figure 3E*). These findings suggest that changes in APA of *Sin3b* mRNA during granule cell differentiation may affect the abundance of the Sin3-HDAC complex, which may in turn affect the exit of granule cells from the cell cycle during development. Another example is evident in cTag PAPERCLIP analysis of APA in the *Rbmx* gene, which encodes a protein required for proper brain development in zebrafish (*Tsend-Ayush et al., 2005*). Whereas both the short and the long isoform of *Rbmx* are expressed in granule cell progenitors, only the long isoform is expressed in differentiated granule cells (*Figure 3E*). The long isoform is predicted to be degraded via nonsense-mediated decay (*Aken et al., 2016*), which may contribute to a decrease in abundance of the *Rbmx* transcript that we observed in differentiated granule cells (*Figure 3E*).

To more systematically corroborate these cTag-PAPERCLIP data, we sequenced RNA isolated from FACS-sorted GFP-positive granule cells from P0 and P21 *Atoh1-Cre; PABP^cTag* mice. The total number of cTag-PAPERCLIP reads per gene (i.e. a measure of the total abundance of 3'UTR isoforms) was highly correlated to the density of RNA-seq reads per gene (RPKM, averaged over all exons of a gene) (R = 0.56, *Figure 3—figure supplement 1*). Moreover, 73% of genes that significantly changed 3'UTR isoform expression during granule cell development as measured by cTag-PAPERCLIP data also changed in the same direction as measured by RNA-seq. RNA sequencing data for the genes mentioned above (*Kpnb1, Cdk7, Sin3b* and *Rbmx*) are shown in *Figure 3—figure supplement 2A and B*. While the pattern of RNA sequencing reads covering the 3'UTRs was consistent with the cTag-PAPERCLIP data, RNA sequencing cannot pinpoint the exact 3'UTR ends. To further validate the differences in 3'UTR isoform abundance for *Kpnb1, Cdk7, Sin3b* and *Rbmx* we performed RT- qPCR on RNA from density gradient-purified granule cells from P0 and P21 cerebella (*Hatten, 1985*). The RT – qPCR data were consistent with cTag-PAPERCLIP data (*Figure 3—figure supplement 3*).

Genes that expressed different 3'UTR isoforms in progenitor and differentiated granule cells were significantly enriched for genes involved in regulating neuritogenesis and dendritic branching (*Supplementary file 6* and *Figure 3F*, which shows the first six most enriched relevant functional categories). These genes include *Nrcam* (*Sakurai, 2012*), which has been shown to regulate axon growth; *Pds5b*, which has been shown to regulate neuronal migration and axon growth (*Zhang et al., 2007*) and *Dclk1*, which has been shown to regulate dendritic vesicle transport and

dendrite development (*Lipka et al., 2016*). That these functions are enriched suggests that APA plays a role in defining the morphological changes between precursor and differentiated granule cells.

Regulatory elements in non-coding 3'UTRs regulate mRNA stability and its association with ribosomes (*Tian and Manley, 2017*). For example, it has been shown that 3'UTR extensions acquired during brain development harbor functional, Argonaute-bound miRNA target sites (*Hwang et al., 2016*; *Miura et al., 2013*). We therefore asked whether changes in APA affected mRNA abundance and its association with ribosomes during granule cell development. We analyzed the abundance of ribosome-associated mRNAs from a recent TRAP study on developing cerebellar granule cells (*Zhu et al., 2016*). Acquiring a longer or shorter 3'UTR during development was not globally correlated to a change in ribosome-associated mRNA abundance (*Figure 3—figure supplement 4*), which is broadly consistent with previous work (*Gruber et al., 2014*; *Gupta et al., 2014*). The lack of a global trend suggests that if APA affects mRNA abundance, it does so in either direction, depending on the gene. Furthermore, we identified transcripts that show significant lengthening of 3'UTRs during development and harbor Argonaute-bound miRNA target sites in the extended 3'UTRs (Argonaute CLIP data from whole cortex from *Moore et al., 2015*). We found that these transcripts, as a group, are not significantly downregulated during development (*Figure 3—figure supplement 4*). This finding underscores that multiple mechanisms likely affect the abundance of most mRNAs besides miRNA targeting.

Finally, we asked whether changes in APA during granule cell development contribute to the differences in APA between mature Purkinje and granule cells. ~ 15% of transcripts with two poly(A) sites that significantly increased distal poly(A) site usage during granule cell development were significantly shorter in Purkinje cells (overlap $p=6.85*10^{-8}$, hypergeometric test, *Figure 4A*). This significant overlap suggests the increase in the use of distal poly(A) sites is a developmentally controlled process to derive neuron-type-specific 3'UTR regulation of mRNA transcripts in the cerebellar granule cells. Examples of genes that showed greater expression of the long 3'UTR isoform in differentiated granule cells compared to proliferating granule cell precursors and Purkinje cells include (*Figure 4B*) *Mbd2*, which codes for methyl CpG-binding domain protein two and has been shown to regulate the proliferation of olfactory receptor neuron precursors (*Macdonald et al., 2010*), *Tmem57*, which codes for macoilin, and has been shown to regulate neuronal excitability (*Arellano-Carbajal et al., 2011*), and *Memo1*, which we describe in more detail below.

## *Memo1* regulates granule cell proliferation and its expression is developmentally regulated by APA and miR-124

Little is known about the physiological consequences of altered 3'UTR length during development (*Boutet et al., 2012*; *Pinto et al., 2011*). To better understand these functional consequences, we focused on APA in *Memo1*, a gene that changed 3'UTR isoform expression during granule cell development. *Memo1* has been implicated in promoting cell motility (*MacDonald et al., 2014*) and in the proliferation of breast cancer cells (*Sorokin and Chen, 2013*), processes which are analogous to events regulated during granule cell development. Granule cell precursors express only the short 3'UTR isoform of *Memo1*, whereas differentiated granule cells express both the short and a long, previously unannotated, isoform ending 14 kb farther downstream (*Figure 5A*). Both the proximal and the distal poly(A) signals of *Memo1*, ending the short and the long 3'UTRs, respectively, are highly conserved across placental mammals, and both isoforms are detectable in human brain RNA (data from [*Jaffe et al., 2015*]). Although they are most salient in cTag-PAPERCLIP data, the presence of both isoforms is consistent with RNA-sequencing data from FACS-sorted granule cells as well as with published TRAP-Seq data (*Mellén et al., 2012*) (*Figure 5—figure supplement 1A*). Interestingly, the overall expression of *Memo1* – as measured by cTag-PAPERCLIP – significantly decreased as granule cells differentiate (*Figure 5B*). We therefore hypothesized that *Memo1* plays a role in proliferating granule cell precursors, when its expression is highest. We addressed this question in the cerebellum of E18.5 *Memo1* knockout mice (*Van Otterloo et al., 2016*), given that these mice die after birth. We observed that the cerebella of *Memo1* knockout mice lacked the typical pattern of fissures, i.e. furrows between which the cerebellar lobes will form. At E18.5, control cerebella had five fissures, whereas *Memo1* knockout cerebella had only one fissure (*Figure 5—figure supplement 1B*). Given that the formation of cerebellar fissures is thought to be driven by the proliferation of granule cells (*Sudarov and Joyner, 2007*), we quantified the number of mitotic cells in the

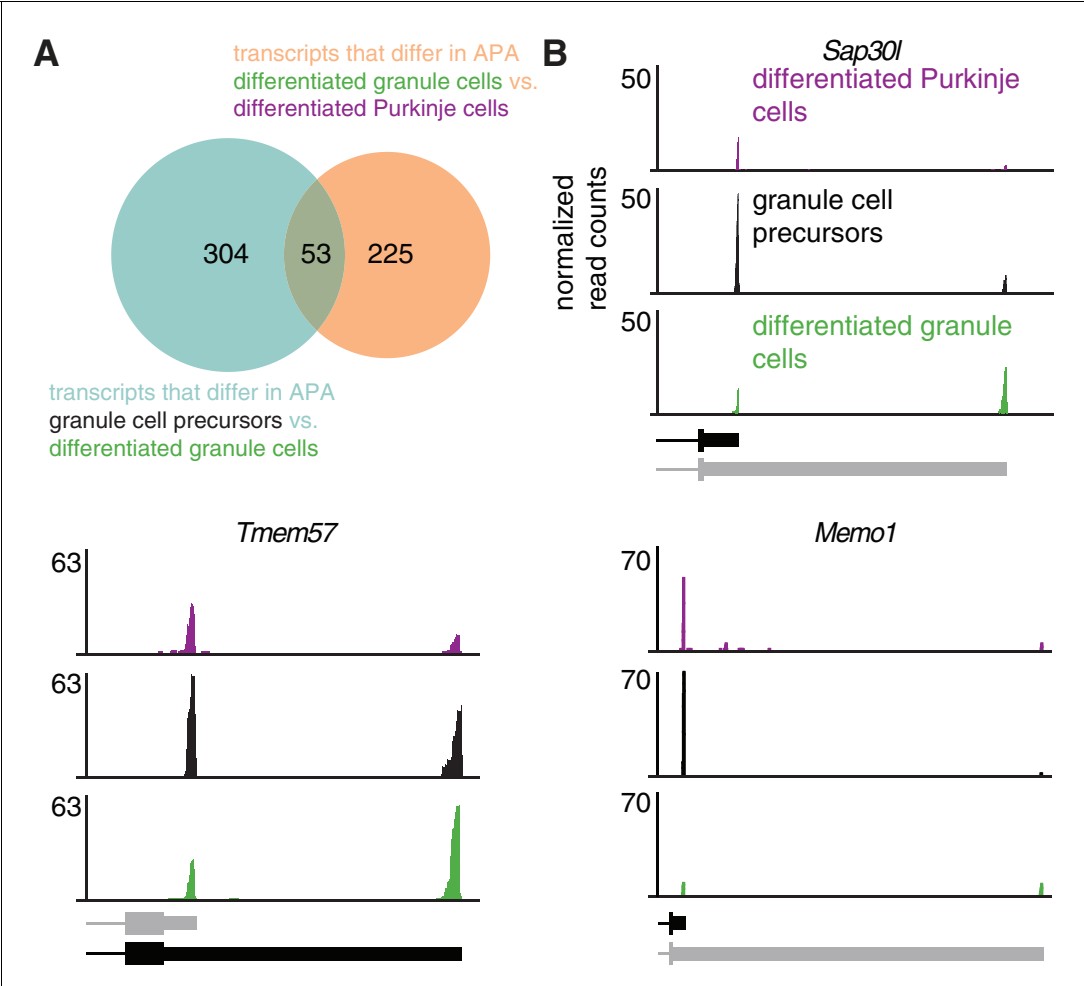

**Figure 4.** Increase in expression of distal 3'UTR isoforms during development is cell-type specific. (**A**) Overlap between genes that exhibit a significant shift towards distal 3'UTR isoform expression during granule cell development (blue circle) and genes that express significantly more of the distal 3'UTR isoforms in granule cells compared to Purkinje cells (orange circle). $p=6.85*10^{-8}$, hypergeometric test. (**B**) cTag-PAPERCLIP data for three genes that show a significant shift towards distal 3'UTR isoform expression during granule cell development and a difference in 3'UTR isoform expression between Purkinje and granule cells. Black bars represent isoforms annotated by Ensembl, gray ones are (predicted) novel isoforms discovered by cTag-PAPERCLIP. The data represents an average of three replicates for adult Purkinje cells and granule cell precursors and four replicates for differentiated granule cells.

DOI: https://doi.org/10.7554/eLife.34042.013

proliferating external granule layer (using anti-phospho-H3 antibody) and found significantly fewer mitotic cells in the external granule layer of *Memo1* knockout mice (*Figure 5C* and S4B). These data suggest that *Memo1* is required for the proliferation of granule cells, consistent with its role in the proliferation of breast cancer cells (*Sorokin and Chen, 2013*).

To identify a potential functional role for the long Memo1 3'UTR acquired during granule cell development, we analyzed the short and the long *Memo1* 3'UTRs for known regulatory sequence motifs (using RegRNA software [*Chang et al., 2013*]). From this analysis, we identified a highly conserved miR-124 target site on the long 3'UTR that was absent in the short 3'UTR (*Figure 5A*). This miR-124 target site was also bound by Argonaute in the mouse cortex, as evidenced by Argonaute CLIP (*Moore et al., 2015*) (*Figure 5A*), suggesting that this target site is functional in the mouse brain. miR-124 is a neuron-specific miRNA that has been shown to be upregulated during differentiation of neural stem cells in the subventricular zone and is required for cell cycle exit and differentiation of neuroblasts (*Cheng et al., 2009*). Thus, we asked whether miR-124 expression also increases during granule cell differentiation. Indeed, we found that miR-124 expression increased between

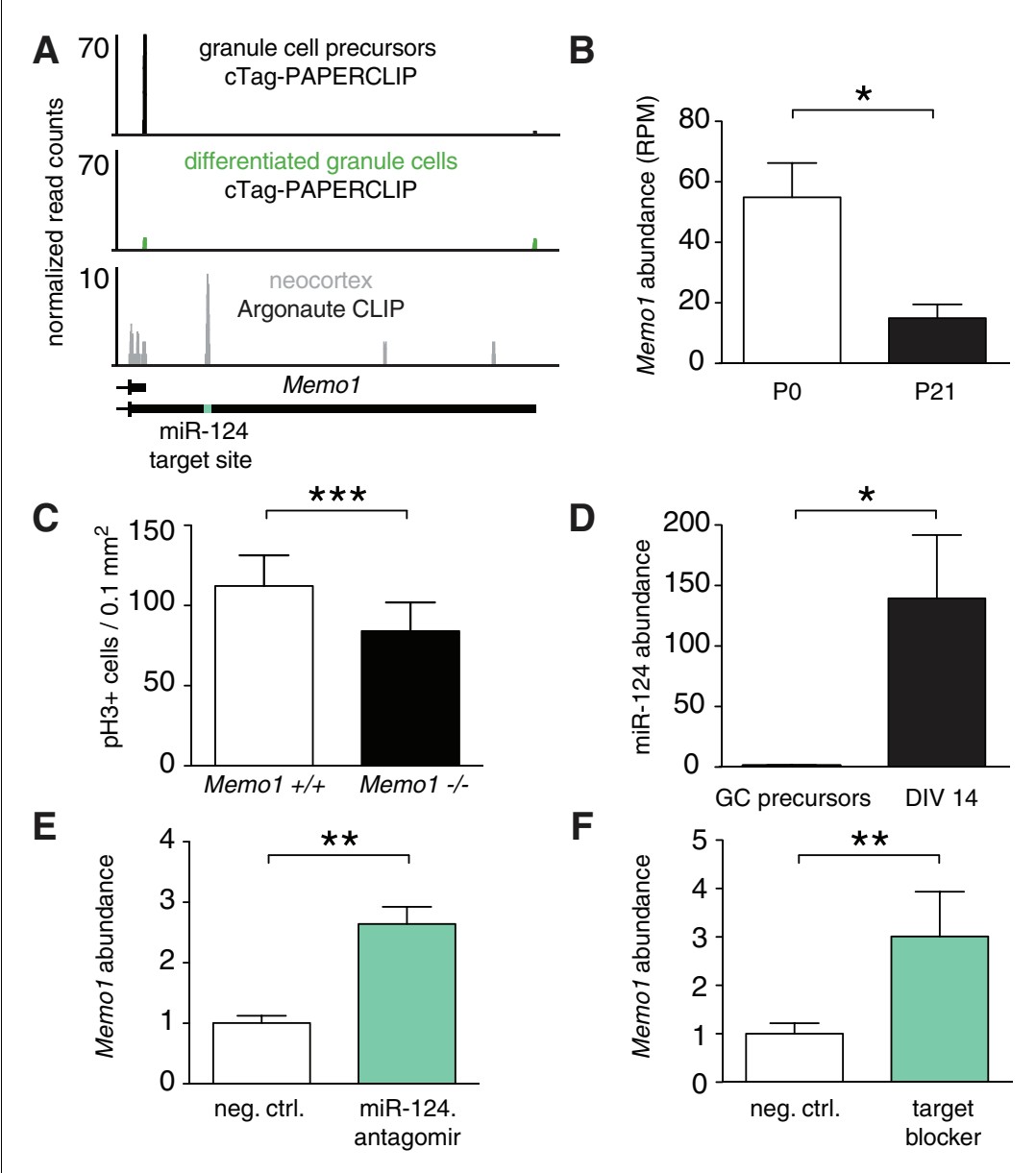

**Figure 5.** *Memo1* expression is developmentally regulated by miR-124 and APA. (**A**) cTag-PAPERCLIP reads on *Memo1* 3'UTR in P0 and P21 granule cells. The location of miR-124 target site is indicated by the green bar. The data represents and average of three cTag-PAPERCLIP replicates and twelve replicates of Argonaute CLIP. (**B**) Expression of *Memo1* gene as determined by cTag-PAPERCLIP. The y-axis represents the number of total unique reads per gene normalized to sequencing depth (RPM – reads per million). Two-tailed t-test. N = 3 for each time point. p-value<0.031 Error bars: standard error. (**C**) Density of phospho-H3 positive (i.e. mitotic) cells in the external granule layer of E18.5 cerebella of wild-type and *Memo1* knockout mice. Two-way ANOVA, N = 3, p-value<0.0001 (see Materials and methods). Error bars: standard error. (**D**) Relative abundance of miR-124 in purified granule cell precursors (GC precursors), after 14 days in culture (DIV 14, two-tailed t-test, N = 3 for each time point, p-value<0.010). Error bars: standard error. (**E**) Relative abundance of long *Memo1* isoform in cultured primary granule cells after treatment with scrambled control (neg. ctrl.) and miR-124 antagomir. Two tailed t-test. N = 2 for both conditions, p-value<0.003. Error bars: standard error. (**F**) Relative abundance of long *Memo1* isoform in cultured primary granule cells after treatment with scrambled control (neg. ctrl.) and *Memo1* miR-124 target site blocker oligonucleotide (target blocker). Two tailed t-test. N = 3 for both conditions. p-value<0.006. Error bars: standard error.

DOI: https://doi.org/10.7554/eLife.34042.014

The following figure supplements are available for figure 5:

**Figure supplement 1.** *Memo1* changes APA during granule cell development and regulates granule cell precursor proliferation.
DOI: https://doi.org/10.7554/eLife.34042.015

**Figure supplement 2.** A model for *Memo1* post-transcriptional regulation during development.

*Figure 5 continued on next page*

*Figure 5 continued*

DOI: https://doi.org/10.7554/eLife.34042.016

purified granule cell precursors and these same precursors differentiated in vitro for 2 weeks, when they morphologically resemble differentiated granule cells (*Figure 5D*). In addition, in situ hybridization experiments have shown high levels of miR-124 in the granule layer of adult mouse cerebella (*Pena et al., 2009*). These results demonstrate that as the expression of miR-124 increases, *Memo1* expression decreases, suggesting a potential role for miR-124 in downregulating *Memo1* expression during granule cell development.

To test whether miR-124 regulates the stability of *Memo1* via its target site on the 3'UTR long isoform, we blocked miR-124 from binding to its targets by transfecting cultured primary granule cells with an antisense oligonucleotide against miR-124. As expected if miR-124 decreases *Memo1* abundance, the *Memo1* long isoform was upregulated under these conditions compared to a scrambled control (*Figure 5E*, antagomir). We further corroborated this miR-124-*Memo1* interaction by blocking the miR-124 target site on *Memo1* with an antisense oligonucleotide (*Figure 5F*, target blocker). These data demonstrate that miR-124 downregulates the *Memo1* long isoform.

Together, these results show that *Memo1* regulates the proliferation of granule cell precursors and cerebellar foliation. During granule cell development, *Memo1* acquires a 3'UTR extension that harbors a miR-124 target site. This extension of the *Memo1* 3'UTR provides a mechanism for *Memo1* downregulation during development (*Figure 5—figure supplement 2*).

## Discussion

Here, we studied differences in APA between specific cell types in vivo using cTag-PAPERCLIP. cTag-PAPERCLIP, unlike other approaches to cell-type specific RNA sequencing (TRAP-seq, RNA-sequencing of purified cells) pinpoints 3'UTR ends exactly (*Figure 2—figure supplement 1*). Compared to RNA-sequencing of purified cells, cTag-PAPERCLIP enables 3'UTR profiling in intact tissue, which avoids effects on gene expression during lengthy cell purification. We recently demonstrated that the expression levels of microglia activation markers and immediate-early genes were significantly lower in the cTag-PAPERCLIP data compared with expression levels obtained by sequencing of RNA from FACS-sorted cells (*Hwang et al., 2017*). Such differences may be accounted for by a number of variables introduced by cell purification schemes that CLIP avoids by crosslinking RNA-protein complexes in vivo, including antibody binding to cell surface markers, disruption of native cell to cell interactions (*Clayton and Darnell, 1983*), or triggering of other cellular stress responses. In addition, cTag-PAPERCLIP enables RNA purification from cellular processes that might get lost during purification. While TRAP-seq avoids some of these problems, it is restricted to analysis of ribosome-bound transcripts while cTag-PAPERCLIP is not, and we have shown that cTag-PAPERCLIP is more selective at purifying cell type specific mRNA from complex tissue (*Figure 1*).

Previous studies have shown that APA can differ between tissues (*Lianoglou et al., 2013*; *Miura et al., 2013*; *Smibert et al., 2012*; *Zhang et al., 2005*) and that 3'UTRs tend to lengthen during embryonic development (*Ji et al., 2009*; *Ulitsky et al., 2012*). Cellular properties have typically been thought of as being controlled at the level of transcription (*Jessell, 2000*), but our results suggest that post-transcriptional RNA regulation, such as APA, may also contribute to the distinct morphological and physiological properties of cerebellar Purkinje and granule cells, as well as to the different properties of precursor and differentiated granule cells.

We found many differences in APA between precursor and differentiated granule neurons, particularly in genes involved in neuronal development (*Figure 3*). These APA switches may regulate the developmental transition between proliferating granule cell precursors to differentiated granule neurons. For example, *Memo1* is required for granule cell proliferation. As granule cell precursors differentiate, *Memo1* increases the relative expression of its long 3'UTR isoform, which harbors a functional miR-124 site (*Figure 5*), that in turn contributes to the downregulation of *Memo1*.

In addition, APA switches during development may contribute to the properties of different mature cell types. We found numerous differences in APA between Purkinje and granule neurons of the cerebellum (*Figure 2*), many of which arose during granule cell differentiation (*Figure 4*). Therefore, we hypothesize that previously observed transcript lengthening during brain development

(*Ji et al., 2009*; *Miura et al., 2013*) occurs in different transcripts across cell types to enable cell type specific post-transcriptional regulation in the adult nervous system, which contributes to the functional diversity between neurons. In particular, we found that *Memo1* undergoes APA in a cell type specific way: Purkinje cells predominantly express the short 3'UTR isoform of *Memo1*, whereas granule neurons express a higher proportion of the miR-124-sensitive long 3'UTR isoform. Given that miR-124 is expressed in both Purkinje and granule cells (*He et al., 2012*; *Pena et al., 2009*), the selective use of the distal *Memo1* poly(A) signal allows for *Memo1* mRNA to be specifically downregulated in granule cells (~3 fold lower compared to Purkinje cells). *Memo1* remains expressed at higher levels in Purkinje cells, where it may play a post-mitotic role. This mechanism of cell type-specific post-transcriptional regulation of *Memo1* is similar to the regulation of *Pax3* by miR-206 in conjunction with APA in muscle stem cells. *Pax3* expresses alternative 3'UTR isoforms based on the location of the muscle (limb vs. diaphragm muscle), which leads to a difference in the abundance of *Pax3* mRNA between these locations (*Boutet et al., 2012*). Interestingly, among genes that expressed more of a longer 3'UTR isoform in granule cells compared to Purkinje cells, 63 genes harbored Argonaute binding sites in extended 3'UTRs and were expressed at lower levels in granule cells compared to Purkinje cells and thus are potentially downregulated by miRNAs (*Supplementary file 8*). Therefore, there are potentially more genes like *Memo1* and *Pax3* that exhibit APA-dependent miRNA-mediated downregulation in specific cell types.

The factors that drive the differences in APA between the cell types profiled here remain unidentified. One possibility is that the overall levels of polyadenylation factors affect polyadenylation site selection. The overall expression levels of polyadenylation factors have been shown to decrease during cell differentiation, as 3'UTRs lengthen, and increase during dedifferentiation, as 3'UTRs shorten (*Elkon et al., 2012*; *Ji and Tian, 2009*). Indeed, polyadenylation factors as a group (*Shi et al., 2009*) were slightly (but not significantly) downregulated (17% median decrease) between granule cell precursors and differentiated granule cells. These observations suggest that an overall decrease in the levels of the core polyadenylation machinery may contribute to the overall lengthening of 3'UTRs during granule cell development. Whether additional levels of APA regulation, other than the overall levels of polyadenylation factors, exist to modulate APA in the genes discussed here will need to be investigated in the future.

Finally, an especially interesting group of transcripts that show differential APA across cell types are those that differ in CDS. We found a number of APA instances that control the inclusion of known protein domains and protein modifications both during in vivo neuronal development and across neuronal classes. As with 3'UTR-APA, these transcripts tend to be longer in differentiated granule cells compared to proliferating precursors, which is consistent with data from MCF10A cell line (*Elkon et al., 2012*), where proliferating cells tend to express isoforms with shorter CDS compared to non-proliferating, serum-starved cells. This finding suggests that there are common regulatory mechanisms affecting both 3'UTR and CDS length during neuronal differentiation, which result in fine-tuning of both the quantity and quality of cell-specific transcripts produced in different kinds of neurons.

# Materials and methods

**Key resources table**

| Reagent type (species) or resource | Designation | Source or reference | Identifiers | Additional information |
|---|---|---|---|---|
| Gene (*Mus musculus*) | *Memo1* | NA | Entrez gene ID: 76890, *Memo1* short isoform RefSeq ID: NM_133771.2 | |
| Genetic reagent (*M. musculus*) | *Neurod1-Cre* | The Jackson Laboratory | 028364 | |
| Genetic reagent (*M. musculus*) | *Pcp2-Cre* | The Jackson Laboratory | 004146 | |
| Genetic reagent (*M. musculus*) | *Atoh1-Cre* | The Jackson Laboratory | 011104 | |

*Continued on next page*

*Continued*

| Reagent type (species) or resource | Designation | Source or reference | Identifiers | Additional information |
|---|---|---|---|---|
| Genetic reagent (*M. musculus*) | *Pabpc1-cTag* | *Hwang et al. (2017)* | PMID: 28910620 | |
| Genetic reagent (*M. musculus*) | (Tg(*Neurod1-Egfp-L10a*) | *Heiman et al. (2008)* | PMID: 19013281 | |
| Genetic reagent (*M. musculus*) | (Tg(*Pcp2-Egfp-L10a*) | *Heiman et al. (2008)* | PMID: 19013281 | |
| Genetic reagent (*M. musculus*) | *Memo1* knockout | *Van Otterloo et al., 2016* | PMID: 26746790 | *Memo1 ki/ki* |
| Antibody | anti-GFP (for cTag-PAPERCLIP) | *Heiman et al. (2008)* | 19F7 and 19C8 | mouse monoclonal |
| Antibody | anti-Calb1 | Santa Cruz | sc-7691 | goat polyclonal, (1:250) |
| Antibody | anti-GFP (for IF) | Aves Labs | GFP-1020 | chicken polyclonal, (1:1000) |
| Antibody | anti-pH3 | Cell Signaling | 9701 | rabbit polyclonal, (1:100) |
| Antibody | Alexa 488, 555 and 647 secondaries | Thermo Fisher | | (1:1000) |
| Sequence-based reagent | miR-124 antagomir | Exiqon | 4102200–121 | mmu-miR-124–3 p inhibitor 3'-fluorescein labeled, (50 nM) |
| Sequence-based reagent | antagomir negative control | Exiqon | 199006–100 | Negative control A, (50 nM) |
| Sequence-based reagent | *Memo1* miR-124 target blocker | Exiqon | 1999993 | Custom miRCURY LNA Power Inhibitor, (50 nM) |
| Sequence-based reagent | target blocker negative control | Exiqon | 199006–111 | Negative control A, (50 nM) |
| Commercial assay or kit | TruSeq RNA Library Prep Kit | Illumina | RS-122–2002 | For preparing RNA-seq libraries |

All primers are listed in **Supplementary file 7**.

## Mice

Animals were maintained in an AAALAC-approved animal facility and all procedures were performed in accordance with IACUC guidelines (protocol number 17013). *Pcp2/L7-Cre*, *Neurod1-Cre* and *Atoh1-Cre* mice were obtained from The Jackson Laboratory (catalog numbers: 004146, 028364 and 011104). *Memo1* mutant mice were generated using ES-cells from the European Conditional Mouse Mutagenesis (EUCOMM) repository as described previously (*Van Otterloo et al., 2016*). Mouse studies were allocated by genotype, and were not blinded.

## Immunohistochemistry

Adult mice used for immunostaining (*Neurod1-Cre; Pabpc1^{cTag}*, *Pcp2-Cre; Pabpc1^{cTag}* and *Atoh1-Cre; Pabpc1^{cTag}* mice) were perfused with PBS and 4% paraformaldehyde (PFA) and fixed in 4% PFA/PBS at 4°C overnight. Heads from E18.5 (*Memo1* mutant mice and WT littermates) and cerebella from P0 mice (*Atoh1-Cre; Pabpc1^{cTag}* mice) were fixed in 4% PFA/PBS at 4°C overnight without perfusion. Fixed heads/cerebella were then incubated in 15% sucrose/PBS followed by 30% sucrose/PBS at 4°C and embedded in O.C.T. compound (Sakura Finetek, Torrance, CA). Adult cerebella were sliced into 30 μm thick sections; E18.5 and P0 cerebella were sliced to 80 μm thick sections on a cryostat (CM3050S, Leica, Germany). The sections were then washed three times with PBS at room temperature (RT), incubated with 0.2% Triton X-100/PBS at RT, blocked with 1.5% normal donkey serum (NDS)/PBS at RT, and incubated overnight at 4°C with primary antibodies in 1.5% NDS/PBS followed by incubation with Alexa 488 and 555 conjugated donkey secondary antibodies (Thermo Fisher, Waltham, MA, 1:1000 dilution) in 1.5% NDS/PBS. Images of immunostained sections were taken using BZ-X700 (Keyence, Japan) microscope. We used the following primary antibodies for immunohistochemistry: anti-Calb1 (Santa Cruz, Dallas, TX, catalog number: sc-7691, 1:250 dilution) anti-GFP (Aves Labs, Tigard, OR, catalog number: GFP-1020, 1:1000 dilution), anti-phospho histone H3 (Cell Signaling Technology, Danvers, MA, catalog number: 9701, 1:100 dilution). Validation of

PABPC1-GFP expression was done on one mouse for each genotype, in at least two technical replicates. The number of phospho-histone H3 positive cells per area of external granule layer was quantified using Photoshop by manually counting phospho-histone H3 positive cells (using Count Tool) and dividing the number with the total external granule layer area. Quantification of phospho-histone H3 positive cells was performed on three *Memo1* knockout mice and three WT mice (from three different litters, knockout and WT mice were littermates), each in four technical replicates (i.e. four sections from the middle of the cerebellum). In this experiment, we observed that the experimental batch (also corresponding to litter) was a significant source of variation, so we performed two-way ANOVA instead of t-test to determine statistical significance of the difference in phospho-H3 staining between WT and *Memo1* KO mice. We used GraphPad Prism5 to perform the calculations.

## cTag-PAPERCLIP

The PAPERCLIP procedure was performed as previously described (*Hwang et al., 2016*). Mouse monoclonal anti-GFP clones 19F7 and 19C8 (*Heiman et al., 2008*) were used for immunoprecipitation. Individual cTag-PAPERCLIP libraries were multiplexed and sequenced on MiSeq (Illumina, San Diego, CA) to obtain 75-nt single-end reads. cTag-PAPERCLIP was performed in four replicates on P56 granule cells, in three replicates on P56 Purkinje cells and in three replicates on developing granule cells (P0 and P21). Due to low amount of RNA obtained by cTag-PAPERCLIP, we pooled 2 mice per replicate for cTag-PAPERCLIP on P21 and P56 granule cells (*Neurod1-Cre; Pabpc1$^{cTag}$* and *Atoh1-Cre; Pabpc1$^{cTag}$* mice), 7 mice per replicate for cTag-PAPERCLIP on P56 Purkinje cells (*Pcp2-Cre; Pabpc1$^{cTag}$* mice) and 10 mice per replicate for cTag-PAPERCLIP on P0 granule cell precursors (*Atoh1-Cre; Pabpc1$^{cTag}$* mice).

## Analysis of cTag-PAPERCLIP data

The processing of raw reads was performed using the CIMS software package as previously described (*Moore et al., 2014*). Raw reads were filtered based on quality score. Filtered reads with the exact sequence were collapsed into one. Poly(A) sequence at the 3′ end was then trimmed using CutAdapt (*Martin, 2011*). Only reads that are at least 25-nt in length were mapped to mm10 reference genome. Mapping was performed using Novoalign (Novocraft, Malaysia) without trimming. Reads mapping to the same genomic positions without distinct barcodes were further collapsed into a single read as previously described (*Moore et al., 2014*). CIMS software package was then used to cluster overlapping collapsed reads from all biological replicates (for each condition) and to determine the number of reads in each cluster. To define a list of high confidence clusters we used the following criteria: clusters had to contain reads from all replicates in each condition, the number of reads in a cluster had to be at least 10% of gene total, clusters had to be located within 20 kb from 3′ ends annotated by RefSeq, clusters that were located upstream of a stretch of 6 or more adenines were excluded due to potential internal priming of the reverse transcription primer. To identify differences in APA between different conditions, Fisher's exact test was performed comparing the ratio between reads in a particular cluster and the sum of reads in all other clusters in a gene in the two conditions. The output p-values were adjusted for multiple hypotheses testing using the Benjamini-Hochberg method. To determine which genes are displaying 3′UTR vs. CDS-APA, we assigned cTag-PAPERCLIP clusters to the nearest Ensembl-annotated 3′UTR using closestBed from the BEDtools suite (*Quinlan and Hall, 2010*). If the start sites of the assigned 3′UTRs for the same gene differed, we assigned the gene as displaying CDS-APA. We manually inspected the CDS-APA gene list and removed a small number of genes that we were not able to unequivocally assign to either 3′UTR-APA or CDS-APA.

To determine overlap between Ensembl 3′UTR ends and cTag-PAPERCLIP clusters, the 3′UTR ends of Ensembl transcripts (Ensembl 89) were extended 10 nucleotides upstream from the end and overlapped with cTag-PAPERCLIP clusters using IRanges and GenomicRanges packages (*Lawrence et al., 2013*).

To compare cTag-PAPERCLIP data with TRAP-Seq data, TRAP-Seq data were mapped to mm10 with Novoalign using default parameters. RPKM values of Ensembl transcripts from TRAP-Seq were then compared to the total number of reads from cTag-PAPERCLIP for the same transcripts.

Protein domains for CDS-APA analysis were obtained manually from Ensembl genome browser (www.ensembl.org). For each alternative transcript, we inspected whether they harbor any known protein domain from protein domain databases (Superfamily, SMART, Pfam and Prosite). Protein modifications were downloaded from Uniprot (*The UniProt Consortium, 2017*) and mapped to mm10 using backlocate software (http://lindenb.github.io/jvarkit/). Genomic coordinates of protein modifications were then overlapped with alternative 3'UTR isoforms to find those that are unique to the long isoform.

To analyze the relationship between 3'UTR isoform expression changes and the abundance of ribosome-associated RNA between P0 and P21 granule cells, microarray data from P0 and P21 granule cells (*Zhu et al., 2016*) were processed using gcrma (*Irizarry and Wu, 2017*) and affy (*Gautier et al., 2004*) software packages, and logFC values were calculated using limma (*Ritchie et al., 2015*) package. To identify 3'UTR isoforms that acquire additional Argonaute binding sites, we overlapped the extended regions of 3'UTRs (i.e. regions between proximal and distal cTag-PAPERCLIP clusters) with Argonaute CLIP clusters (data from whole cortex from *Moore et al., 2015*) using findOverlaps from IRanges package (*Lawrence et al., 2013*). We only considered Argonaute CLIP clusters that contained at least five unique CLIP tags.

## Gene ontology analysis

Gene ontology analysis was performed on genes showing significant changes in 3'UTR isoform expression using Ingenuity Pathway Analysis software (Qiagen, Germany) Core Analysis tool. Background was determined as all genes with cTag-PAPERCLIP clusters expressed in Purkinje and granule cells or P0 and P21 granule cells. 'User Dataset' was chosen as 'Reference Set.' Genes with significant differences in APA were given 'Expr Other' value of 1, all other genes were given 'Expr Other' value of 0. Cutoff was chosen as 1. We focused on functions related to 'Molecular and Cellular Functions' and 'Physiological System Development and Function' that are overrepresented in the set of genes that show significant changes in 3'UTR isoform expression between Purkinje and granule cells and during development of granule cells. For the *Figures 2D* and *3E* we manually removed categories that were not relevant or contained very similar gene sets (Supplementary Tables contain all significant functional categories).

## FACS sorting and RNA sequencing

GFP-positive granule cells from P0 and P21 *Atoh1-Cre; Pabpc1^cTag^* mice were sorted using FACSAria II (BD, Franklin Lakes, NJ). Three P0 and two P21 mice were used for this experiment. Post-sort analysis showed that the sorted samples contained 99% of GFP-positive cells. RNA was isolated from sorted cells using Roche High Pure RNA isolation kit as per manufacturer's instructions. RNA sequencing library was prepared using TruSeq kit from Illumina, following manufacturer's instructions. Individual RNA-seq libraries were multiplexed and sequenced on HiSeq (Illumina, San Diego, CA) to obtain 100-nt paired-end reads. RNA-sequencing reads were mapped to the mouse genome (mm10) using Novoalign (Novocraft, Malaysia), reads overlapping genes were counted using HTSeq-count (*Anders et al., 2015*). The total number of cTag-PAPERCLIP reads per transcript was then compared to the total number of RNA-sequencing reads per transcript.

## Translating-ribosome Affinity Purification (TRAP)

mRNA from translating polysomes was purified from mouse cerebella at P56 as previously described (*Heiman et al., 2008*; *Zhu et al., 2016*). Heterozygous transgenic mice carrying EGFP-tagged ribosomal protein L10a (*Tg(Neurod1-Egfp-L10a)*) and (*Tg(Pcp2-Egfp-L10a)*) (*Heiman et al., 2008*) were used to purify mRNA from granule and Purkinje cells, respectively.

## Granule cell purification and culture

Granule cells were prepared as described (*Hatten, 1985*). Briefly, cerebella were dissected away from the brains of P6 mice. After the pial layer was peeled away, the tissue was treated with trypsin for 5 min at 37°C and triturated into a single-cell suspension using fine bore Pasteur pipettes. The suspension was layered on a discontinuous Percoll gradient and separated by centrifugation. The small cell fraction was isolated, and granule cells were further enriched by panning on tissue culture treated plastic dishes. The resulting cultures routinely contain greater than 95% of cells of the

granule cell lineage (*Hatten, 1985*). For the experiments using miR-124 antagomir and *Memo1* target blocker, granule cells were plated in serum-free granule cell medium on Poly-D lysine coated 6-well plates and transfected with 50 nM miR-124–3p antagomir or *Memo1* target blocker (Exiqon, Woburn, MA, custom miRCURY LNA Power Inhibitor, final concentration 50 nM for both) and appropriate negative control using Lipofectamine RNAiMAX (Thermo Fisher, Waltham, MA). Cells were then let to differentiate (4 days) and RNA was isolated using TRIzol (Thermo Fisher, Waltham, MA) followed by High Pure RNA Isolation Kit (Roche, Switzerland) following manufacturer's instructions. The experiment with miR-124–3p antagomir was performed in two biological replicates (each biological replicate was performed in two technical replicates) and the experiment with *Memo1* target blocker was performed in three biological replicates (each biological replicate was performed in two technical replicates). Biological replicate is defined as an independent experiment (an independent isolation of primary granule cells from P6 pups followed by in vitro differentiation and RNA isolation). Technical replicate is defined as an independent quantitative PCR reaction.

To purify mRNA for cTag-PAPERCLIP validation by qPCR, we purified granule cells from P0 cerebella as described above. We followed the same protocol to purify granule cells from P21 cerebella, except that we treated cerebella with papain for 30 min at 37°C.

## Reverse transcription and quantitative PCR

Quantification of *Memo1* long isoform in granule cell culture was performed by reverse transcription using SuperScript III reverse transcriptase (Thermo Fisher, Waltham, MA) and quantitative PCR using FastStart SYBR Green Master mix (Roche, Switzerland). Primers to amplify long *Memo1* isoform are listed in *Supplementary file 7*. We used the following cycling parameters: 95°C for 10 min. followed by 40 cycles of 95°C for 15 s., 58°C for 30 s., 72°C for 20 s. Fold changes and p-values were calculated using Bio-Rad CFX Maestro Software. To quantify miR-124 we isolated RNA from granule cells using High Pure RNA Isolation Kit (Roche) and reverse transcribed it using miScript II RT Kit (Qiagen, Germany) following manufacturer's instructions. We then performed quantitative PCR using FastStart SYBR Green Master mix (Roche, Switzerland) using the following parameters: 94°C for 15 min, 40 cycles (15 s at 94°C, 30 s at 55°C and 30 s at 70°C). Primers to amplify miR-124 are listed in *Supplementary file 7*.

For qPCR validation experiments, mRNA (purified by TRAP from P56 Purkinje and granule cells and extracted from P0 and P21 granule cells upon purification using Percoll gradient as described above) was reverse transcribed using iScript cDNA Synthesis Kit (Bio-Rad, Hercules, CA) following manufacturer's instructions. qPCR was performed using FastStart SYBR Green Master mix (Roche, Switzerland). Primers to amplify the short vs. long 3'UTR isoforms are listed in *Supplementary file 7*. We used the following cycling parameters: 95°C for 10 min. followed by 40 cycles of 95°C for 15 s., 58°C for 30 s., 72°C for 20 s. We then calculated the change in ratio between the long 3'UTR isoform vs. total mRNA for 3'UTR-APA and short 3'UTR isoform only for CDS-APA genes using Prism5 (GraphPad Software, La Jolla, CA). We calculated the same ratio using cTag-PAPERCLIP data.

## Acknowledgements

We thank Trevor Williams at the University of Colorado for assistance with the *Memo1* null mice, Yin Fang for help with granule cell purification, Chris Park for advice on data analysis and providing us with Argonaute CLIP data, Mariko Kobayashi, Joseph Luna and Jonathan Green for critical reading of the manuscript. We acknowledge Xiaodong Zhu and Keisha John who purified TRAP RNA used in this study. We also thank the Rockefeller University Flow Cytometry Facility. This work was in part supported by grants from the National Institutes of Health (NS034389 and NS081706) and Simons Foundation (SFARI 240432) to RBD and by grant K99DE026823 from the NIDCR to EVO. RBD is an Investigator of the Howard Hughes Medical Institute.

## Additional information

### Funding

| Funder | Grant reference number | Author |
| --- | --- | --- |
| National Institutes of Health | NS034389 | Robert B Darnell |
| Howard Hughes Medical Institute | | Robert B Darnell |
| Simons Foundation | SFARI 240432 | Robert B Darnell |
| National Institute of Dental and Craniofacial Research | K99DE026823 | Eric Van Otterloo |
| National Institutes of Health | NS081706 | Robert B Darnell |
| National Institutes of Health | NS097404 | Robert B Darnell |
| National Institutes of Health | 1UM1HG008901 | Robert B Darnell |

The funders had no role in study design, data collection and interpretation, or the decision to submit the work for publication.

### Author contributions

Saša Jereb, Conceptualization, Data curation, Software, Formal analysis, Investigation, Visualization, Methodology, Writing—original draft, Writing—review and editing; Hun-Way Hwang, Resources, Writing—review and editing; Eric Van Otterloo, Eve-Ellen Govek, John J Fak, Investigation, Writing—review and editing; Yuan Yuan, Methodology, Writing—review and editing; Mary E Hatten, Resources, Supervision, Project administration, Writing—review and editing; Robert B Darnell, Conceptualization, Resources, Supervision, Funding acquisition, Writing—review and editing

### Author ORCIDs

Saša Jereb http://orcid.org/0000-0001-6862-4475

Yuan Yuan http://orcid.org/0000-0002-2718-8301

Mary E Hatten http://orcid.org/0000-0001-9059-660X

Robert B Darnell http://orcid.org/0000-0002-5134-8088

### Ethics

Animal experimentation: Animals were maintained in an AAALAC-approved animal facility and all procedures were performed in accordance with IACUC guidelines (protocol number 17013).

### Decision letter and Author response

Decision letter https://doi.org/10.7554/eLife.34042.035
Author response https://doi.org/10.7554/eLife.34042.036

## Additional files

### Supplementary files

• Source Code 1. *Supplementary file 4* - source code 1. Source code to identify genes that exhibit significantly different 3'UTR isoform expression between granule cell precursors and mature granule cells
DOI: https://doi.org/10.7554/eLife.34042.017

• Supplementary file 1. Genes differing in 3'UTR isoform expression between Purkinje and granule cells
DOI: https://doi.org/10.7554/eLife.34042.018

• Supplementary file 2. Genes differing in CDS length between Purkinje and granule cells
DOI: https://doi.org/10.7554/eLife.34042.019

• Supplementary file 3. Functional gene categories enriched among genes that differ in 3'UTR isoform expression between Purkinje and granule cells
DOI: https://doi.org/10.7554/eLife.34042.020

• Supplementary file 4. Genes differing in 3'UTR isoform expression between granule cell precursors and mature granule cells
DOI: https://doi.org/10.7554/eLife.34042.021

• Supplementary file 5. Genes differing in CDS length between granule cell precursors and mature granule cells
DOI: https://doi.org/10.7554/eLife.34042.022

• Supplementary file 6. Functional gene categories enriched among genes that differ in 3'UTR isoform expression between Purkinje and granule cells
DOI: https://doi.org/10.7554/eLife.34042.023

• Supplementary file 7. List of PCR primers used in the study
DOI: https://doi.org/10.7554/eLife.34042.024

• Supplementary file 8. Transcripts that express more of the longer 3'UTR isoform in granule cells and are downregulated compared to Purkinje cells
DOI: https://doi.org/10.7554/eLife.34042.025

• Transparent reporting form
DOI: https://doi.org/10.7554/eLife.34042.026

## Major datasets

The following dataset was generated:

| Author(s) | Year | Dataset title | Dataset URL | Database, license, and accessibility information |
|---|---|---|---|---|
| Saša Jereb | 2017 | Differential 3' Processing of Specific Transcripts Expands Regulatory and Protein Diversity Across Neuronal Cell Types | https://www.ncbi.nlm.nih.gov/geo/query/acc.cgi?acc=GSE108480 | Publicly available at the NCBI Gene Expression Omnibus (accession no: GSE108480). |

The following previously published datasets were used:

| Author(s) | Year | Dataset title | Dataset URL | Database, license, and accessibility information |
|---|---|---|---|---|
| Mellén M, Ayata P, Dewell S, Kriaucionis S, Heintz N | 2013 | MeCP2 binds to 5hmC enriched within active genes and accessible chromatin in the nervous system | https://www.ncbi.nlm.nih.gov/geo/query/acc.cgi?acc=GSE42880 | Publicly available at the NCBI Gene Expression Omnibus (accession no: GSE42880). |
| Zhu X, Girardo D, Govek EE, John K, Mellén M, Tamayo P, Mesirov JP, Hatten ME | 2015 | Role of Tet1/3 Genes and Chromatin Remodeling Genes in Cerebellar Circuit Formation | https://www.ncbi.nlm.nih.gov/geo/query/acc.cgi?acc=GSE74402 | Publicly available at the NCBI Gene Expression Omnibus (accession no: GSE74402). |

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
