## [Decision Letter]

Thank you for submitting your article "Differential 3' Processing of Specific Transcripts Expands Regulatory and Protein Diversity Across Neuronal Cell Types" for consideration by *eLife*. Your article has been reviewed by three peer reviewers, and the evaluation has been overseen by a Reviewing Editor and a Senior Editor. The following individual involved in review of your submission has agreed to reveal his identity: Eugene Mekeyev (Reviewer #3).

The reviewers have discussed the reviews with one another and the Reviewing Editor has drafted this decision to help you prepare a revised submission.

Summary:

In this study, Darnell and colleagues use a recently developed technique, cTap-PAPERCLIP, to compare transcriptome-wide usage of alternative polyadenylation (APA) sites between two types of cerebellar neurons, Purkinje and granule cell, as well as between mature granule cells and their proliferating precursors. Interestingly, mRNA isoforms in the three cell types are shown to have readily distinguishable 3' ends affecting the 3'UTRs and, occasionally, the protein-coding sequences. This extends previously published analyses of global trends toward lengthening of 3'UTRs during development in general and brain development in particular. To begin addressing possible biological consequences of the cell type-specific 3'-terminal dynamics, the authors focused on a cell motility and proliferation factor, Memo1, whose mRNA gains a longer 3'UTR during differentiation of granule cell precursors into neurons. The authors show that the extended 3'UTR acquires a functional binding site for the microRNA miR-124, which is naturally up-regulated in differentiating granule cell neurons. The authors show that the interaction between miR-124 and its cognate 3'UTR sequence dampens Memo1 expression in granule cell. Finally, evidence is provided for a role of Memo1 in granule cell precursor proliferation in vivo. All in all, this is a nice study that should be of considerable interest to *eLife* readers.

Essential revisions:

1) The main finding of this work is that distinct types of neurons and their precursors may express mRNA isoforms with surprisingly diverse 3' ends. However, this conclusion is drawn entirely based on bioinformatics analyses of cTap-PAPERCLIP, TRAP-seq and RNA-seq data. The authors should validate at least some of the cell type-specific events reported in their manuscript using RT-qPCR. This would be a relatively straightforward experiment since the authors have access to fluorescently labeled cells that can be isolated by FACS as described in the "FACS sorting and RNA sequencing" section. Likewise, some of the APA events need to be validated by RT-qPCR as well.

2) The authors should show all the data mentioned in the manuscript as either main or supplemental panels. This applies, for example, to subsection “cTag-PAPERCLIP Maps mRNA 3’UTR Ends in Specific Cells in vivo*”* ("We found a high correlation between the two methods for both Purkinje cells (R = 0.68, Figure 1E) and granule cells (R=0.7, data not shown)") and also subsection “cTag-PAPERCLIP Maps mRNA 3’UTR Ends in Specific Cells in vivo*”* ("Known marker genes for each cell type were among the most highly expressed genes in the Purkinje cell (Figure 1E) and granule cell datasets (data not shown)"). In addition, the authors discuss some of their analyses without showing the data (Discussion section: "Interestingly, among genes that expressed more of a longer 3'UTR isoform in granule cells compared to Purkinje cells, 150 genes were expressed at lower levels in granule cells compared to Purkinje cells and thus are potentially downregulated by miRNAs."). This should be remedied – ideally by showing the results.

3) The mechanism of Memo1 regulation by APA is well presented. But alternative models need to be considered. For example, decrease in transcription leading to downregulation of short 3'UTR isoform and concomitant stabilization of long 3'UTR can also result in observed APA and expression changes. Can this be ruled out? What is the direct evidence for its APA regulation? It would also be helpful if the authors can present a cartoon to summarize the model.

---

## [Author Response]

Essential revisions:1) The main finding of this work is that distinct types of neurons and their precursors may express mRNA isoforms with surprisingly diverse 3' ends. However, this conclusion is drawn entirely based on bioinformatics analyses of cTap-PAPERCLIP, TRAP-seq and RNA-seq data. The authors should validate at least some of the cell type-specific events reported in their manuscript using RT-qPCR. This would be a relatively straightforward experiment since the authors have access to fluorescently labeled cells that can be isolated by FACS as described in the "FACS sorting and RNA sequencing" section. Likewise, some of the APA events need to be validated by RT-qPCR as well.

We appreciate the reviewer’s point, and we have now performed RT-qPCR validations for 8 genes shown in Figure 2 and Figure 3.

We initially attempted to purify sufficient numbers of fluorescently labeled cells after FACS to undertake robust RT-qPCR per the reviewer’s suggestion, but these experiments were tricky, and our yield was in the low ng range, so we opted to use alternative methods to purify granule cell RNA rather than re-deriving mice which would be timely and costly.

To validate 3’UTR isoform differences between granule and Purkinje cells shown in Figure 2, we obtained TRAP-purified RNA from granule and Purkinje cells (same RNA that was used for microarray studies in Zhu et al., 2016). For validation of differential 3’UTR isoform expression of genes in Figure 3 we now purified granule cells using a density gradient centrifugation method (Hatten, 1985) from P0 and P21 mice. We show data from these validation experiments as figure supplements to Figure 2 and Figure 3.

Differences in 3’UTR isoform expression ratio between the cell types were consistent between RT-qPCR and cTag-PAPERCLIP data for all genes shown in Figure 2 and Figure 3. Interestingly, for all genes shown in Figure 2, cTag-PAPERCLIP showed more pronounced differences in 3’UTR isoform expression between the two cell types compared to RT-qPCR on TRAP-purified RNA, presumably because of the more selective purification of RNA (Figure 2—figure supplement 2).

We believe these experiments should fully address the reviewers request.

2) The authors should show all the data mentioned in the manuscript as either main or supplemental panels. This applies, for example, to subsection “cTag-PAPERCLIP Maps mRNA 3’UTR Ends in Specific Cells in vivo” ("We found a high correlation between the two methods for both Purkinje cells (R = 0.68, Figure 1E) and granule cells (R=0.7, data not shown)") and also subsection “cTag-PAPERCLIP Maps mRNA 3’UTR Ends in Specific Cells in vivo” ("Known marker genes for each cell type were among the most highly expressed genes in the Purkinje cell (Figure 1E) and granule cell datasets (data not shown)"). In addition, the authors discuss some of their analyses without showing the data (Discussion section: "Interestingly, among genes that expressed more of a longer 3'UTR isoform in granule cells compared to Purkinje cells, 150 genes were expressed at lower levels in granule cells compared to Purkinje cells and thus are potentially downregulated by miRNAs."). This should be remedied – ideally by showing the results.

We appreciate the reviewer’s points. We have now added a supplementary panel (Figure 1—figure supplement 1) to show the data comparing TRAP-seq with cTag-PAPERCLIP for granule cells. We also made a very minor change in the analysis of the Purkinje cell data in Figure 1E (we filtered out genes that showed very low expression in TRAP-seq data) to make the analysis consistent with our analysis of granule cell data shown in Figure 1—figure supplement 1.

Regarding the results we mentioned in the Discussion section, we have now updated our analysis and focused on those genes that, in addition to expressing a significantly longer 3’UTR isoform in granule cells, are both downregulated and bound by Argonaute. We list all of these genes in Supplementary file 8.

3) The mechanism of Memo1 regulation by APA is well presented. But alternative models need to be considered. For example, decrease in transcription leading to downregulation of short 3'UTR isoform and concomitant stabilization of long 3'UTR can also result in observed APA and expression changes. Can this be ruled out? What is the direct evidence for its APA regulation? It would also be helpful if the authors can present a cartoon to summarize the model.

We appreciate the reviewer’s suggestion that we consider alternative models more carefully. We considered selective transcriptional downregulation of the short transcript; given that both isoforms have the same transcription start site, we would consider this a less likely possibility than that proposed.

However, we addressed the possibility that both isoforms are transcriptionally down-regulated, and the long isoform is concomitantly stabilized. To address the question of transcriptional downregulation we performed qPCR on *Memo1* pre-mRNA and mature mRNA on RNA from Percoll-purified P0 cells and on RNA from whole cerebellum from P21 mice (which consists of ~90% granule cells). We used whole cerebellum from P21 mice because we were unable to isolate enough RNA to amplify Memo1 pre-mRNA from purified P21 granule cells (due to very low cell and RNA yields of the Percoll-based purification). We observed that *Memo1* pre-mRNA was indeed downregulated during development (~3-fold), suggesting the possibility that there is some transcriptional downregulation of the gene; however, we found a much larger downregulation of mature *Memo1* mRNA (~23-fold), suggesting that the major contribution to the downregulation of *Memo1* during developmentresults from a post-transcriptional mechanism (see Author response image 1). Given that we found that long *Memo1* 3’UTR contains a miR-124 site that we find experimentally is functional, we argue that the conversion of *Memo1* to an unstable isoform via alternative polyadenylation is likely to explain a significant component of the effect observed.

We thank the reviewer again and have now added a cartoon that clarifies and summarizes this model (Figure 5—figure supplement 2).

**Author response image 1. respfig1:** *Memo1* mature mRNA and pre-mRNA expression. RT-qPCR data showing relative abundance of *Memo1* mature mRNA and pre-mRNA in purified granule cells from P0 cerebella and whole P21 cerebella.N=3, Error bars: SEM. *Memo1* mRNA and pre-mRNA fold changes between the two conditions (P0 and P21) are 23.8 and 3.2 fold, respectively. P-value for difference in log2(FC P0/P21) = 0.0292.